# Transformer-based Live Update Generation
# for Soccer Matches from Microblog Posts

**Masashi Oshika**       **Kosuke Yamada**       **Ryohei Sasano**       **Koichi Takeda**

Graduate School of Informatics, Nagoya University, Japan

`{oshika.masashi.f6,yamada.kosuke.v1}@s.mail.nagoya-u.ac.jp`
`{sasano,takedasu}@i.nagoya-u.ac.jp`

## Abstract

It has been known to be difficult to generate adequate sports updates from a sequence of vast amounts of diverse live tweets, although the live sports viewing experience with tweets is gaining the popularity. In this paper, we focus on soccer matches and work on building a system to generate live updates for soccer matches from tweets so that users can instantly grasp a match's progress and enjoy the excitement of the match from raw tweets. Our proposed system is based on a large pre-trained language model and incorporates a mechanism to control the number of updates and a mechanism to reduce the redundancy of duplicate and similar updates.

## 1 Introduction

When a sports match is broadcast, Twitter users often enjoy sharing the status of the match or their opinions. For example, during a televised soccer match, many users post tweets that include information about how goals were scored or their thoughts on a certain play, and it is possible to roughly understand a match's progress by reading these tweets. However, because of the diverse nature of tweets, ranging from informative to purely emotive content, it can be challenging to quickly grasp a match's progress. In this study, we focus on soccer matches and work on building a system to generate live sports updates from tweets so that users can instantly grasp a match's progress.

Generation of live sports updates from tweets can be considered a type of multi-document summarization (McKeown and Radev, 1995). Much research exists on multi-document summarization, including recent studies that aim to generate high-quality summarization by capturing the hierarchical structure of documents (Fabbri et al., 2019; Jin et al., 2020). More recently, pre-trained language models such as BART and PRIMERA have also been used (Pasunuru et al., 2021; Xiao et al.,

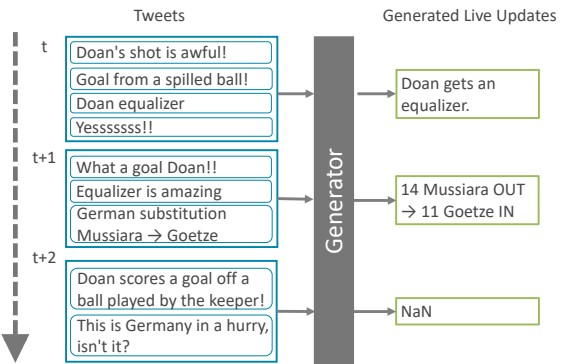

Figure 1: Overview of live sports update generation. Tweets in the figure show the timestamped sequence of tweets containing hashtags associated with a match such as #WorldCup or #UCL. Hashtags are excluded when tweets are input into the generator.

2022). However, these methods are not easy to apply to this task because they do not account for time series data. Time-series-aware update generation methods can be divided into two approaches depending on the generation timing. One approach determines the timing according to the number of posts per unit time (Nichols et al., 2012; Kubo et al., 2013; Tagawa and Shimada, 2016). While this approach can accurately capture main events, it tends to miss less significant events that have fewer related tweets. Another approach, which generates updates at regular intervals (Dusart et al., 2021), can generate exhaustive updates but tends to produce redundancies by mentioning the same event at different times after it occurs.

In this paper, we build a pre-trained language model-based system to generate live updates for soccer matches. In particular, we propose to incorporate a mechanism to control the number of updates and a mechanism to mitigate redundancy in the system. Figure 1 shows an overview of our system. It uses the pre-trained language model Text-to-Text Transfer Transformer (T5) (Raffel et al., 2020), for which the inputs are tweets related to a specific match and the outputs are updates at cer-

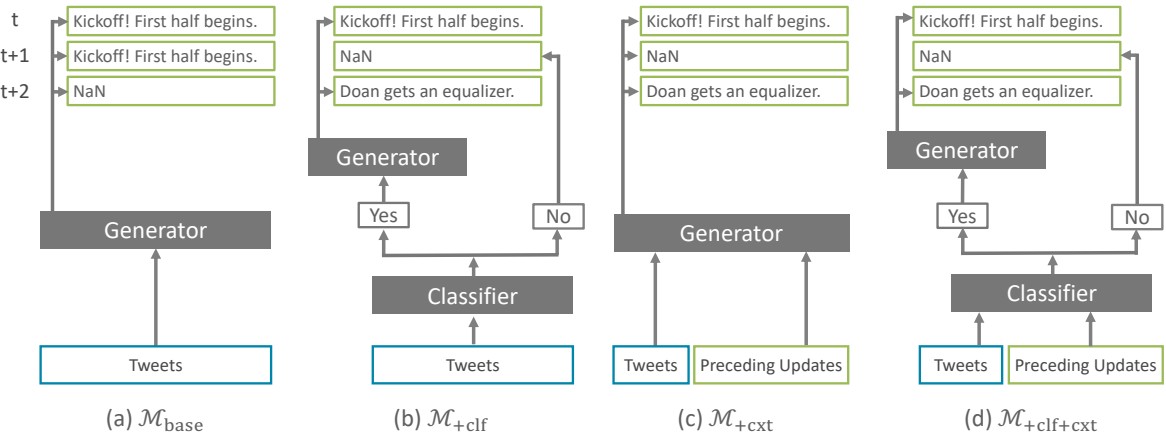

Figure 2: Architecture of the baseline and proposed models.

tain times. If there is no appropriate update, "NaN" is output. To control the number of updates, it uses a binary classifier to determine whether an update should be generated at each time, and to mitigate redundancy, it leverages preceding updates.

## 2 Proposed Method

Our system is based on an abstractive model using T5. We also apply the approach that generates the update every minute. As a preliminary experiment regarding the generation timing, we tried another approach, which is detecting events such as goals or substitutions by monitoring the increasing number of tweets and generating updates at those detected points. However, we confirmed that some events such as close shot or good defensive plays could not be detected. For this reason, our models typically generate one update per minute and output "NaN" if they determine that it is better not to output any update.

Figure 2 (a) shows the architecture of our baseline model ($\mathcal{M}_{\mathrm{base}}$). It generates updates every minute without referring to preceding updates; thus, updates may be redundant if an event is mentioned repeatedly over a long period. In addition, "NaN" is more frequently generated unless a highly probable word is predicted at the beginning of the output, which may result in too many "NaN." To address these problems, in this study, we propose a model that incorporates a classifier to determine whether to output "NaN" and a mechanism to mitigate redundancy by incorporating preceding updates.

### 2.1 Classifier

To limit the number of "NaN" outputs, we introduce a model that separates the classifier of whether to output "NaN" from the generator ($\mathcal{M}_{\mathrm{+clf}}$), as

shown in Figure 2 (b). Specifically, before generating an update, this model performs binary classification of whether to generate an update (*yes*) or not (*no*) from input tweets. If the decision is *yes*, then an update is generated by the generator; otherwise, "NaN" is output. We use T5 for the classifier so that we can construct the model using only a pre-trained T5 although we can use pre-trained models based on a Transformer encoder, such as BERT (Devlin et al., 2019) or RoBERTa (Liu et al., 2019). [1] Note that the weights of these T5 models are not shared.

### 2.2 Incorporation of Preceding Updates

To avoid generating redundant updates, we introduce a model in which the preceding updates are added as input in addition to tweets as context information ($\mathcal{M}_{\mathrm{+cxt}}$), as shown in Figure 2 (c). Previous studies have shown considerable improvement of performance on NLP tasks by using prompts that combine input text with its related text (Liu and Chen, 2021; Liu et al., 2022; Wang et al., 2022). These studies led us to the idea of incorporating the preceding context for generating updates as follows. The preceding updates are combined using the T5 special token and then input to the model in the form of "*tweets* `<extra_id_0>` *preceding updates*." Since the classifier and this generation mechanism are independent, we can define the fourth model that combines them ($\mathcal{M}_{\mathrm{+clf+cxt}}$) as shown in Figure 2 (d).

### 2.3 Fine-tuning Setup

Both the classifier and the generator are constructed by fine-tuning a pre-trained T5. The classifiers used

---

[1] We experimented with pre-trained language model other than T5 as classifiers, such as BERT, but could not observe major differences in performance.

in $\mathcal{M}_{+\text{clf}}$ and $\mathcal{M}_{+\text{clf}+\text{cxt}}$ are fine-tuned to output either *yes* or *no* by inputting only tweets, or tweets and their previous updates, respectively. The generators used in $\mathcal{M}_{\text{base}}$ and $\mathcal{M}_{+\text{cxt}}$ are fine-tuned so that the generator can generate an update or "NaN." The generators used in $\mathcal{M}_{+\text{clf}}$ and $\mathcal{M}_{+\text{clf}+\text{cxt}}$ are fine-tuned to always generate updates, using only the time periods when updates are present. In $\mathcal{M}_{+\text{cxt}}$ and $\mathcal{M}_{+\text{clf}+\text{cxt}}$ that use preceding updates, updates included in the training data are used as the preceding updates for fine-tuning, and the automatically generated updates are used during the generation.

## 3 Experiment

To confirm the proposed method's effectiveness, we compared the performance of the baseline model, three proposed models, and two oracle models.

### 3.1 Settings

**Dataset** We first collected data for 68 weeks of the J-League, the Japanese professional soccer league, from 2021 and 2022, comprising a total of 612 matches. Of these, we used 83 matches with more than 3200 tweets between one hour before the match and one hour after the match. In total, we collected 8,502 updates and 324,835 tweets. We then created five subsets based on match and performed five-fold cross-validation with training:development:test = 3:1:1. We used the training set to fine-tune the model, the development set to find the best model, and the test set to evaluate the model. We used the Twitter API[2] to collect tweets. We obtained information on game times and opposing teams from the official J-League website, and we created the query for the API consisting of the game time to be collected, and manually determined hashtags such as #grampus and #fmarinos related to that match. We collected tweets using the queries and pre-processed the tweets to remove URLs and hashtags in the tweets.

We used the text updates published by the official J-League website[3] as the reference updates. Some updates were related to results up to the previous week or lifetime achievements. Since it is considered to be out of scope to generate such updates from live tweets, we excluded these updates from reference updates using manually created rules.

**Models** In addition to the proposed models, we built two oracle models. The first model is $\text{Orcl}_{\text{extr}}$, which is an oracle extractive model that measured the similarity between the reference update and all the tweets at the corresponding time and extracted the tweets with the highest percentage of matching words. This method is an oracle version of Dusart et al. (2021)'s method, and we used it to investigate the upper limit of the achievable performance of the extractive approach. The second model is $\text{Orcl}_{+\text{clf}+\text{cxt}}$, which uses the reference updates as the preceding context in the generation phase. The model is based on $\mathcal{M}_{+\text{clf}+\text{cxt}}$. It enables us to investigate the oracle performance that the proposed method could achieve. These models only generated updates at times when the reference updates exist to ensure that the numbers of generated updates matched those of reference updates.

$\mathcal{M}_{+\text{cxt}}$ and $\mathcal{M}_{+\text{clf}+\text{cxt}}$, which incorporate preceding updates, used the updates of the last 4 minutes before each time, and all methods used tweets posted up to 3 minutes after each time. All models are based on Japanese pre-trained T5[4] on Hugging Face Hub. All models were fine-tuned with three different random seeds to build three versions of the models, and their average was used as the evaluation score.

**Metrics** We evaluated the methods by comparing the generated updates with the reference updates. Because similar events could occur multiple times, an evaluation that assesses word agreement throughout the match could result in an unfairly high score. To address this, we evaluated the similarity of strings after mapping them on the timeline.

ROUGE (Lin, 2004) is a commonly used evaluation metric for text generation, and alignment-based ROUGE (Martschat and Markert, 2017) is an extension of ROUGE that was designed for timeline summarization. In this study, although we accounted for time differences for the generated updates, we did not apply weights to the time differences. Our evaluation was based on the number of overlapping n-grams, defined as **aligned n-grams**, between the reference and generated updates. Specifically, dynamic programming is applied to maximize the aligned n-grams while allowing for a one-minute difference. More details are provided in Appendix A.

---

[2] https://developer.twitter.com/ja
[3] https://www.jleague.jp

[4] https://huggingface.co/megagonlabs/
t5-base-japanese-web

| | # of reference unigrams | # of generated unigrams | # of aligned unigrams | Precision | Recall | F1-score |
|---|---|---|---|---|---|---|
| $\mathcal{M}_{\text{base}}$ | 164,245 | 49,088 | 19,585 | **0.401** | 0.118 | 0.182 |
| $\mathcal{M}_{+\text{clf}}$ | 164,245 | 129,387 | 42,199 | 0.326 | 0.255 | 0.286 |
| $\mathcal{M}_{+\text{cxt}}$ | 164,245 | 107,049 | 41,639 | 0.389 | 0.251 | **0.305** |
| $\mathcal{M}_{+\text{clf}+\text{cxt}}$ | 164,245 | 137,072 | 43,074 | 0.318 | **0.260** | 0.284 |
| $\text{Orcl}_{\text{extr}}$ | 164,245 | 109,046 | 33,159 | 0.304 | 0.200 | 0.241 |
| $\text{Orcl}_{+\text{clf}+\text{cxt}}$ | 164,245 | 134,598 | 51,487 | 0.382 | 0.311 | 0.343 |

Table 1: Experimental results when unigrams are used for evaluation.

| | Reference updates | Generated updates ($\mathcal{M}_{\text{base}}$) | Generated updates ($\mathcal{M}_{+\text{cxt}}$) |
|---|---|---|---|
| t | NaN | 1 minute of extra time. | 1 minute of extra time. |
| t+1 | 1 minute of extra time. | 1 minute of extra time. | The first half ended with the game tied at 0-0. |
| t+2 | The first half ended with the game tied at 0-0. | The first half ended with the game tied at 0-0. | Second-harlf kickoff, YokohamaFM ball. No halftime substitutions for either team. |
| t+3 | Second-harlf kickoff, YokohamaFM ball. 10 MJunio OUT → 16 Fujita IN. 19 Ogashiwa OUT→23 Koroki IN | NaN | NaN |

Table 2: Examples of reference and generated updates.

## 3.2 Main Results

Table 1 shows the results when unigrams are used for evaluation.[5] As expected, $\mathcal{M}_{\text{base}}$ generated many "NaN," because it generated fewer unigrams as compared to the reference unigrams. In contrast, for $\mathcal{M}_{+\text{clf}}$, the number of generated unigrams was close to the number of the reference unigrams, suggesting that the classifier enabled more appropriate decisions on whether to generate updates. Comparing $\mathcal{M}_{\text{base}}$ and $\mathcal{M}_{+\text{cxt}}$, we confirmed that the recall was improved considerably while the precision remained the same. This shows that by incorporating the preceding context, the number of updates to be generated can be controlled to some extent, while successfully suppressing redundant output.

Next, by comparing $\mathcal{M}_{+\text{cxt}}$ and $\mathcal{M}_{+\text{clf}+\text{cxt}}$, we can confirm that the introduction of the classifier resulted in a slight improvement in recall, but a significant decrease in precision and consequently in F-score. We speculate that this is due to the following: the model with the classifier is forced to generate updates if the decision of the classifier is *yes*, even when the corresponding set of tweets contains almost no information necessary to generate the reference updates, which significantly decreases the precision and thus the F1-score as well. Overall, $\mathcal{M}_{+\text{cxt}}$ achieved the best scores among the proposed models. We performed paired permutation tests on the differences in F1-scores between $\mathcal{M}_{+\text{cxt}}$ and the other models and found that all differences were significant at the significance level of 0.001.

Lastly, we compared the two oracle models with the proposed models. All of the proposed models scored higher than $\text{Orcl}_{\text{extr}}$. This suggests that the extractive approach has a clear limitation for this task and that it is preferable to apply the abstractive approach for generating live sports updates from tweets. The $\text{Orcl}_{+\text{clf}+\text{cxt}}$ achieved a higher recall while maintaining precision compared to $\mathcal{M}_{+\text{cxt}}$, which achieved the highest F1-score among the proposed methods. However, the recall is only 0.311, which suggests the existence of reference updates that are very difficult to generate from the set of tweets posted in real time.

## 3.3 Analysis

Table 2 lists examples of reference updates and updates generated by $\mathcal{M}_{\text{base}}$ and the best proposed model $\mathcal{M}_{+\text{cxt}}$. Note that the actual updates are in Japanese, but their English translations are provided here for legibility. We can see that while the baseline model generated the same updates repeatedly, redundant updates were suppressed by $\mathcal{M}_{+\text{cxt}}$. Regarding the overall output, however, the generation of redundant updates was not completely suppressed, which remains to be a challenge. In addition, the comparison of the reference updates with those generated by $\mathcal{M}_{+\text{cxt}}$ shows that updates mentioning the same event were output one minute off. Such time discrepancies are not considered a major practical problem, and this result confirms the effectiveness of the evaluation metrics, which allowed for small time differences.

We further conducted manual evaluations to investigate how accurately each model was able to

---

[5]Results with bigrams are provided in Appendix B.

| Event | Metrics | $\mathcal{M}_{\mathrm{base}}$ | $\mathcal{M}_{\mathrm{+clf}}$ | $\mathcal{M}_{\mathrm{+cxt}}$ | $\mathcal{M}_{\mathrm{+clf+cxt}}$ | $\mathrm{Orcl}_{\mathrm{extr}}$ | $\mathrm{Orcl}_{\mathrm{+clf+cxt}}$ |
|---|---|---|---|---|---|---|---|
| Goals | precision | 0.71 / **0.43** | 0.66 / 0.42 | **0.75** / 0.43 | 0.73 / 0.27 | 0.88 / 0.76 | 0.54 / 0.28 |
| | recall | 0.53 / 0.32 | 0.66 / 0.42 | **0.87** / **0.50** | 0.84 / 0.32 | 0.79 / 0.68 | 0.87 / 0.45 |
| | F1-score | 0.61 / 0.36 | 0.66 / 0.42 | **0.80** / **0.46** | 0.78 / 0.29 | 0.83 / 0.72 | 0.67 / 0.34 |
| Player substitutions | precision | 0.71 / 0.36 | 0.80 / 0.33 | **0.86** / 0.29 | 0.75 / **0.38** | 0.97 / 0.82 | 0.76 / 0.30 |
| | recall | 0.11 / 0.05 | 0.13 / 0.05 | 0.19 / 0.06 | **0.26** / **0.13** | 0.74 / 0.63 | 0.37 / 0.15 |
| | F1-score | 0.19 / 0.09 | 0.22 / 0.09 | 0.31 / 0.10 | **0.38** / **0.19** | 0.84 / 0.71 | 0.50 / 0.20 |
| Penalty cards | precision | 0.40 / 0.20 | 0.42 / 0.23 | **0.50** / **0.38** | 0.42 / 0.32 | 0.83 / 0.67 | 0.33 / 0.17 |
| | recall | 0.15 / 0.07 | **0.41** / **0.22** | 0.30 / **0.22** | 0.30 / **0.22** | 0.19 / 0.15 | 0.30 / 0.15 |
| | F1-score | 0.22 / 0.11 | **0.42** / 0.23 | 0.37 / **0.28** | 0.35 / 0.26 | 0.30 / 0.24 | 0.31 / 0.16 |
| Total | precision | 0.65 / 0.37 | 0.61 / 0.34 | **0.73** / **0.38** | 0.67 / 0.32 | 0.94 / 0.79 | 0.58 / 0.27 |
| | recall | 0.21 / 0.12 | 0.30 / 0.17 | 0.37 / **0.19** | **0.40** / **0.19** | 0.66 / 0.56 | 0.48 / 0.22 |
| | F1-score | 0.32 / 0.18 | 0.40 / 0.23 | 0.49 / **0.26** | **0.50** / 0.24 | 0.77 / 0.66 | 0.52 / 0.24 |

Table 3: Manual evaluation results. Each cell lists the results of lenient / strict evaluation.

detect key events in the matches. Specifically, we considered three types of key events: goals, player substitutions, and penalty cards, and manually evaluated whether these events were detected correctly. The evaluation was performed using two criteria: lenient and strict. In the case of lenient evaluation, each event was evaluated using precision, recall, and F1-score, where each event was considered to be correctly detected if the reference updates and generated updates contained the same type of event within 2 minutes. In the case of strict evaluation, each event was considered to be correctly detected only when the scorer was identical for goals, when the type of card and the name of the target player were identical for penalty cards, and when the names of the target players were identical for substitutions.

Table 3 shows the results of the human evaluation on 10 randomly selected matches. Overall, $\mathrm{Orcl}_{\mathrm{extr}}$ achieved a high score, which is natural since $\mathrm{Orcl}_{\mathrm{extr}}$ is a model that selects tweets that are most consistent with the reference updates. Other than the oracle models, $\mathcal{M}_{\mathrm{+cxt}}$ and $\mathcal{M}_{\mathrm{+clf+cxt}}$ achieved relatively high scores. It appears that by considering the context, redundant output can be avoided, resulting in higher key event detection performance. However, the score for the strict evaluation is not high enough, especially when compared to $\mathrm{Orcl}_{\mathrm{extr}}$. This indicates that the generated updates do not include detailed information on key events such as player names, even though the source tweet set in most cases includes such information, and more accurate detection of key events remains as future work.

## 4 Conclusion

In this paper, we have investigated the update generation of soccer matches from tweets. We presented models based on T5 and attempted to incorporate

a classifier to control the number of updates and preceding updates to reduce redundancy. We confirmed that our method can achieve high performance by considering preceding updates. However, the generation of redundant updates was not completely suppressed, and this remains a challenge for implementing an update generation system with higher performance. In the future, we would like to investigate update generation for other sports.

## Limitations

Our experiment has two limitations. First, we have not been able to conduct experiments using models other than T5 in the generator. It is possible that performance could be improved by using a different model. Secondly, the experiment was conducted only on soccer matches in Japanese. The effectiveness towards other languages and other sports is unverified.

## Acknowledgements

This work was partly supported by JSPS KAKENHI Grant Number 21K12012.

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

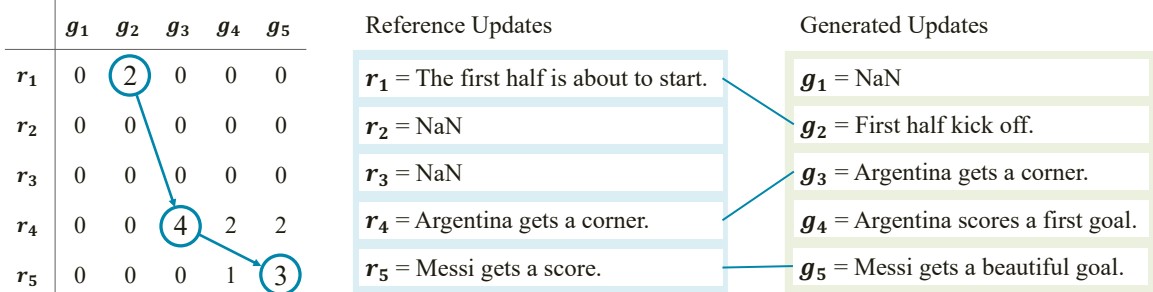

|       | $g_1$ | $g_2$ | $g_3$ | $g_4$ | $g_5$ |
|-------|-------|-------|-------|-------|-------|
| $r_1$ | 0     | ②     | 0     | 0     | 0     |
| $r_2$ | 0     | 0     | 0     | 0     | 0     |
| $r_3$ | 0     | 0     | 0     | 0     | 0     |
| $r_4$ | 0     | 0     | ④     | 2     | 2     |
| $r_5$ | 0     | 0     | 0     | 1     | ③     |

Reference Updates

$r_1$ = The first half is about to start.
$r_2$ = NaN
$r_3$ = NaN
$r_4$ = Argentina gets a corner.
$r_5$ = Messi gets a score.

Generated Updates

$g_1$ = NaN
$g_2$ = First half kick off.
$g_3$ = Argentina gets a corner.
$g_4$ = Argentina scores a first goal.
$g_5$ = Messi gets a beautiful goal.

Figure 3: Example of the evaluation procedure. The matrix stores the number of overlapping n-grams between the $g_i$ and $r_j$. After applying dynamic programming, circles are placed on the reference updates and generated updates, and their paths are connected by arrows. On the right side, examples of reference updates and generated updates are shown, and the aligning ones are connected.

|                          | # of reference bigrams | # of generated bigrams | # of aligned bigrams | Precision | Recall | F1-score |
|--------------------------|------------------------|------------------------|----------------------|-----------|--------|----------|
| $\mathcal{M}_{base}$     | 159,605                | 46,505                 | 8,825                | **0.192** | 0.055  | 0.085    |
| $\mathcal{M}_{+clf}$     | 159,605                | 123,680                | 16,160               | 0.130     | **0.101** | 0.114 |
| $\mathcal{M}_{+cxt}$     | 159,605                | 102,739                | 16,309               | 0.156     | 0.100  | **0.122** |
| $\mathcal{M}_{+clf+cxt}$ | 159,605                | 131,441                | 15,752               | 0.121     | 0.098  | 0.108    |
| $Orcl_{extr}$            | 159,605                | 103,406                | 3,447                | 0.033     | 0.021  | 0.026    |
| $Orcl_{+clf+cxt}$        | 159,605                | 128,958                | 20,764               | 0.161     | 0.130  | 0.143    |

Table 4: Experimental results when bigrams are used for evaluation.

## A  Evaluation Details

The following is the evaluation procedure used in this study and Figure 3 illustrates the evaluation procedure with an example.

1. Define arrays $g$ and $r$ to store the generate updates and reference updates, respectively, and a matrix $S$ to store the mapping scores between updates, $S = [s_{ij}]$.

2. Assign to $s_{ij}$ the number of overlapping n-grams between the $g_i$ and $r_j$, where $i, j$ are indexes of the match time. To prevent mapping between updates with large time differences, assign 0 when $|i - j| > 1$.

3. Apply dynamic programming to $S$ to map a generated update to a reference update so as to maximize the number of overlapping n-grams in the entire match, under the constraint that each update is mapped to at most one reference update. Denote this maximum number as **aligned n-gram**. Evaluate a model in terms of three indices for **aligned n-gram**: the precision, recall, and F1-score.

## B  Experimental Results with Bigrams

Table 4 lists the results of an experiment using bigrams. The scores were generally lower when using unigrams, but the overall tendencies were similar. $\mathcal{M}_{+cxt}$ had the highest F1-score among the proposed models, while $Orcl_{+clf+cxt}$ scored the highest among all six models. $Orcl_{extr}$ had the lowest score, unlike in the case of unigrams. It is considered difficult for extractive methods to generate updates that are very close to the reference updates.