# OpenReview forum: "Transformer-based Live Update Generation for Soccer Matches from Microblog Posts"
_EMNLP/2023/Conference — EMNLP 2023 Main_

### Official Review · Reviewer_23Lj · 2023-08-03

**Soundness:** 4

**Excitement:**

3: Ambivalent: It has merits (e.g., it reports state-of-the-art results, the idea is nice), but there are key weaknesses (e.g., it describes incremental work), and it can significantly benefit from another round of revision. However, I won't object to accepting it if my co-reviewers champion it.

**Paper Topic And Main Contributions:**

In this paper, authors propose a Transformer-based live update generation for soccer matches based on tweets. Specifically, it adopts the T5 model as the backbone to generate live updates. To control the number of updates, it uses a binary classifier to determine whether an update should be generated at each time, and to mitigate redundancy, it leverages preceding updates.

**Reasons To Accept:**

1. The idea of this work is simple and interesting.

**Reasons To Reject:**

1. In the experiments, this paper adopts the number of overlapping unigrams, which lacks persuasiveness.
2. This article is more suitable for demo session.

**Reproducibility:**

4: Could mostly reproduce the results, but there may be some variation because of sample variance or minor variations in their interpretation of the protocol or method.

**Reviewer Confidence:**

4: Quite sure. I tried to check the important points carefully. It's unlikely, though conceivable, that I missed something that should affect my ratings.

---

> ### Author Rebuttal · Authors · 2023-08-28
>
> We sincerely thank you for reviewing our paper.
> >In the experiments, this paper adopts the number of overlapping unigrams, which lacks persuasiveness.
>
> Thank you for your important remarks. In the case of a soccer match update, for example, if both the reference and the generated update contain the same person's name and the word "scored," the generated update is considered the appropriate update in most cases. On the basis of this thinking, we employed an evaluation based on unigrams, and bigrams, as shown in Table 3 in the Appendix. However, we agree with the point that these evaluations are not sufficient. We would like to add a human evaluation of agreement rates for events that are definitely included in the updates, such as scoring, player changes, etc.

---

### Official Review · Reviewer_Yzg3 · 2023-08-05

**Soundness:** 4

**Excitement:**

3: Ambivalent: It has merits (e.g., it reports state-of-the-art results, the idea is nice), but there are key weaknesses (e.g., it describes incremental work), and it can significantly benefit from another round of revision. However, I won't object to accepting it if my co-reviewers champion it.

**Paper Topic And Main Contributions:**

In this paper, the authors built a pre-trained language model-based system to generate live updates for soccer matches.

**Questions For The Authors:**

- The reviewer did not understand how reference updates look like. There should be gap between reference collected from J-League official site and tweets. So, ROUGE may not the best evaluation metric.
- The reviewer is not sure what　"preceding update" is at final.

**Reasons To Accept:**

- The topic "Live update generation" is interesting and useful, and it can be expanded to other topics.
- The paper shows tweets can be good candidates as an information source to replicate the reference.

**Reasons To Reject:**

- The authors did not prepare other baselines in the experiments, so it is difficult to compare the proposed systems' performance.
- The performance on the best method is still low and the difficulty of the task is unclear because there is not an explanation of an upper bound.

**Reproducibility:**

3: Could reproduce the results with some difficulty. The settings of parameters are underspecified or subjectively determined; the training/evaluation data are not widely available.

**Reviewer Confidence:**

4: Quite sure. I tried to check the important points carefully. It's unlikely, though conceivable, that I missed something that should affect my ratings.

---

> ### Author Rebuttal · Authors · 2023-08-28
>
> The authors would like to thank the invaluable questions and comments.
> > The performance on the best method is still low and the difficulty of the task is unclear because there is not an explanation of an upper bound.
>
> We agree that we do not have a very detailed analysis of the difficulty of this task. However, we believe that the low performance of 0.241 for the F1-score of Orcl_extr, which is to select the tweets that will give the highest score, indicates that this is a difficult task and that the score of around 0.3 achieved by the proposed model is relatively high.
> We also believe that experiments based on the Orcl+clf+cxt model, which uses the correct information about whether to generate an update at that time and past updates, are experiments that provide some sort of upper bound.
> > The reviewer did not understand how reference updates look like. There should be gap between reference collected from J-League official site and tweets. So, ROUGE may not the best evaluation metric.
>
> The "reference updates" column in Table 2 shows the English translation of the actual reference updates. In addition, you can find the actual reference update from https://www.jleague.jp, although the data is in Japanese, e.g., https://www.jleague.jp/match/j1/2023/082602/live/#livetxt.
> It is true that there is a gap between references collected from the official J-League website and tweets, so when employing an extractive summarization strategy, ROUGE may not be an appropriate metric for evaluation, but since this study employs an abstractive method, we believe that ROUGE is one of the best evaluation metrics.
> > The reviewer is not sure what　"preceding update" is at final.
>
> In this study, we define "preceding updates" as updates in the three minutes immediately preceding the time at which the update is to be generated. For example, considering the timing of t+3 in Table 2, the reference update from t to t+2, are the preceding updates. When performing fine-tuning, or generating updates with the Orcl+clf+cxt model, this would specifically be "NaN", "1 minute of extra time.", and "The first half ended with the game tied at 0-0."

---

### Official Review · Reviewer_hkC3 · 2023-08-07

**Soundness:** 4

**Excitement:**

3: Ambivalent: It has merits (e.g., it reports state-of-the-art results, the idea is nice), but there are key weaknesses (e.g., it describes incremental work), and it can significantly benefit from another round of revision. However, I won't object to accepting it if my co-reviewers champion it.

**Paper Topic And Main Contributions:**

In this work, the authors focus on the application of generating live sports from tweets by leveraging powerful transformer-based text-to-text models such as T5 models. Strategies are proposed to control the number of updates and mitigate the redundancy in the system. More specifically, a binary classifier is used to determine whether an update should be generated. By considering the preceding updates, the system can further alleviate the redundancy issue. Experimental results on J-League related tweets have shown the effectiveness of the proposed method.

**Questions For The Authors:**

1. In line 105-106, the authors said, “We use T5 for the classifier so that we can construct the model using only a pre-trained T5”. However, in line 110, the authors said, “The weights of these T5 models are not shared”. Then what’s the benefit of only using one architecture?
2. Maybe out of the scope of this short paper, what’s the performance by applying Dusart et al. 2021 to this dataset?


**Reasons To Accept:**

1. The method is easy to understand and the application is quite interesting.
2. Data construction method is useful for the community. The evaluation method is comprehensive. The significance test is conducted.


**Reasons To Reject:**

This work is an application-oriented task where the pretrained language models are applied to a specific task that is underexplored. I will challenge the model contribution for this work but in terms of new application, it is a neat short paper.


**Reproducibility:**

4: Could mostly reproduce the results, but there may be some variation because of sample variance or minor variations in their interpretation of the protocol or method.

**Reviewer Confidence:**

4: Quite sure. I tried to check the important points carefully. It's unlikely, though conceivable, that I missed something that should affect my ratings.

---

> ### Author Rebuttal · Authors · 2023-08-28
>
> We greatly thank you for your time and efforts in reviewing our paper.
> > In line 105-106, the authors said, "We use T5 for the classifier so that we can construct the model using only a pre-trained T5". However, in line 110, the authors said, "The weights of these T5 models are not shared". Then what's the benefit of only using one architecture?
>
> It is true that there is no advantage in using the same pre-trained model for both generators and classifiers in terms of either computational cost or required memory size, but we chose to use the same pre-trained model for both generators and classifiers because of the advantage of having only one language model to be prepared in advance.
>
> > Maybe out of the scope of this short paper, what's the performance by applying Dusart et al. 2021 to this dataset?
>
> We have not checked the performance of applying Dusart et al.'s model, but we can say that Dusart et al.'s model is an extractive model and therefore will have lower performance than Orcl_extr, an extractive oracle model.

---

### Meta-Review · Area_Chair_CYUj · 2023-09-07

**Recommendation:** 4

**Metareview:**

This paper investigates the task of generating live sports updates from tweets and proposes a series of T5-based models that controls the number of updates to be generated to mitigate redundancy. Overall, reviewers are rather positive about this submission and agree that the proposed task / models are interesting and that the content is a good fit for a short paper. On the negative side, reviewers point out that the experiments could be strengthened (more baselines, oracle / upper bound performance evaluation metrics).

---

### Decision · Program_Chairs · 2023-10-07

**Decision:**

Accept-Main

**Comment:**

This paper investigates the task of generating live sports updates from tweets and proposes a series of T5-based models that controls the number of updates to be generated to mitigate redundancy. Overall, reviewers are rather positive about this submission and agree that the proposed task / models are interesting and that the content is a good fit for a short paper. On the negative side, reviewers point out that the experiments could be strengthened (more baselines, oracle / upper bound performance evaluation metrics).